# Effect of Ho Substitution on Magnetic Properties and Microstructure of Nanocrystalline Nd-Pr-Fe-B Alloys

**Caihai Xiao** [1,2,†], **Weiwei Zeng** [2,†], **Yongli Tang** [3], **Cifu Lu** [2], **Renheng Tang** [2], **Zhigang Zheng** [1,*], **Xuefeng Liao** [2] **and Qing Zhou** [2,*]

1   School of Materials Science and Engineering, South China University of Technology, Guangzhou 510640, China
2   Guangdong Provincial Key Laboratory of Rare Earth Development and Application, Institute of Resources Utilization and Rare Earth Development, Guangdong Academy of Sciences, Guangzhou 510640, China
3   Huizhou Feller Magnets Co., Ltd., Huizhou 518055, China
*   Correspondence: mszgzheng@scut.edu.cn (Z.Z.); zqwork@grre.gd.cn (Q.Z.)
†   These authors contributed equally to this work.

**Abstract:** The inevitable thermal demagnetization of magnets at high-temperatures is a key issue for Nd-Fe-B based permanent magnetic materials, especially for electric motors. Here, we report the effect of partially substituting the element Holmium (Ho) on the magnetic properties and microstructure of nanocrystalline melt-spun $[(NdPr)_{1-x}Ho_x]_{14.3}Fe_{76.9}B_{5.9}M_{2.9}$ (x = 0–0.6; M = Co, Cu, Al and Ga) alloys. It shows that Ho can enter into the main phase and significantly enhance the coercivity ($H_{cj}$). A large coercivity of 23.9 kOe is achieved in the x = 0.3 alloy, and the remanent magnetization ($M_r$) remains in balance. The abnormal elevated temperature behavior of $M_r$ is observed in the alloys with a high amount of Ho substitution, in which the $M_r$ of the x = 0.6 alloy increases with rising temperature from 300 K to 375 K owing to the antiparallel coupling between Ho and Fe moments. As a result, the positive value (0.050%/K) of temperature coefficient $\alpha$ of $M_r$ is achieved in the x = 0.6 alloy within the temperature range of 300–400 K, in excess of that of existing Nd-Fe-B magnets. The temperature coefficient $\beta$ of $H_{cj}$ is also improved by Ho substitution, indicating the introduction of Ho in Nd-Fe-B magnets is beneficial for thermal stability. The microstructure observation of x = 0, 0.3 and 0.6 alloys confirmed the grain refinement by Ho substitution, and Ho prefers to remain in the 2:14:1 phase than Nd and Pr. The present finding provides an important reference for the efficient improvement of the thermal stability of Nd-Fe-B-type materials.

**Keywords:** Nd-Fe-B ribbons; Ho-substituted; coercivity; remanent magnetization; thermal stability





## 1. Introduction

Nd-Fe-B-based magnets have been employed as the high-performance permanent magnetic material for numerous applications, such as in traction motors and generators, due to their excellent hard magnetic properties [1,2]. As required for the developing motor market, Nd-Fe-B magnets with superior thermal stability are seriously needed to maintain performance in high temperatures [3]. Enhancing coercivity is considered to be an effective way of withstanding the demagnetizing field at high service temperatures [4,5]. The heavy rare earth (RE) elements Dy and Tb are generally added to Nd-Fe-B magnets to improve their coercivity by the alloying or grain boundary diffusion method, since $Dy_2Fe_{14}B$ and $Tb_2Fe_{14}B$ exhibit higher anisotropy fields ($H_A$) than that of the $Nd_2Fe_{14}B$ compound [6]. However, the addition of Dy or Tb causes a problem regarding the price of the magnets, due to the limited natural abundance and the high cost of Dy/Tb. In the last few years, many efforts have been dedicated to the development of Dy/Tb-free Nd-Fe-B magnets with high coercivity to resist thermal demagnetization [3,7,8]. However, the outcomes are still insufficient.

Holmium (Ho) is one of the less-abundant heavy RE elements in the earth's crust, but its price is relatively low because it is also one of the less-frequently used. Although $Ho_2Fe_{14}B$ only presents a slightly higher $H_A$ (75 kOe) than 73 kOe of $Nd_2Fe_{14}B$, several researchers have indicated that introducing Ho into Nd-Fe-B magnets can dramatically improve the coercivity [9]. A high coercivity of 18 kOe was realized in Dy-free sintered Nd-Fe-B magnets by the intergranular addition of Ho-Fe powder [10]. In addition, the $Ho_2Fe_{14}B$ compound shows a different temperature-dependence behavior of magnetization ($M_s$) from $Nd_2Fe_{14}B$, in which the $M_s$ decreases with decreasing temperature, owing to the antiparallel coupling between Ho and Fe moments [11]. Therefore, the Ho-containing sintered Nd-Fe-B magnets exhibit superior thermal stability to that of Ho-free magnets. For nanocrystalline melt-spun Nd-Fe-B alloys, it is found that 20% Ho substitution can significantly improve both coercivity and high-temperature performance [12]. Recent work on high-abundance Ce-based melt-spun alloys shows that the coercivity of $(Ce_{1-x}Ho_x)_{14}Fe_{80}B_6$ can be dramatically improved from 5 kOe to 17 kOe with 50% Ho addition; meanwhile the Curie temperature $T_c$ and thermal stability are also greatly improved [13]. In addition, the introduction of trace elements has a positive effect on the magnetic properties of Nd-Fe-B magnets. Substituting the element Co for Fe can significantly increase the Curie temperature ($T_c$) of the $Nd_2Fe_{14}B$ phase; thus, Co is usually employed to improve the thermal stability of Nd-Fe-B magnets [14]. Although Al is not favorable to $T_c$, it is found to be beneficial for coercivity by partially substituting Fe for a decrease in domain wall thickness [15]. Adding Ga and Cu can regulate the grain boundary, and results in coercivity enhancement [16,17].

Therefore, Ho has the potential to improve the magnetic properties of Nd-Fe-B magnets. However, until now, there has been no systematic report about the effect of Ho substitution on melt-spun Nd-Fe-B alloys. In this study, the effects of Ho substitution on the phase constitution, magnetic properties and microstructure of the melt-spun (Nd,Pr)-Fe-B alloys were systematically investigated. The magnetic properties at room and high temperature instances of the alloys after Ho addition were discussed in detail. This work provides useful guidelines for enhancing the hard magnetic properties of (Nd,Pr)-Fe-B magnets. More importantly, it offers a practical roadmap for reducing thermal demagnetization.

## 2. Experimental

A series of Ho-added alloy ingots with nominal compositions of $[(Nd_{0.8}Pr_{0.2})_{1-x}Ho_x]_{14.3}Fe_{76.9}B_{5.9}M_{2.9}$ (M = Co, Cu, Al and Ga) (at. %; x = 0–0.6) were prepared by induction melting technique under Ar atmosphere. The starting materials were Nd-Pr, Ho, Fe, Fe-B, Co, Cu, Al and Ga metals with purities higher than 99.9%. Hereafter, the samples are simply labeled as NPHFBM. The nanocrystalline melt-spun ribbons were prepared by melt spinning. The alloys with a total mass of 100 g were melted by induction melting, and the general steps of alloy melting were as follows: alloys were subjected to 1 kW heat preservation for 1 min for preheating, then the power was increased to 7 kW for melting; after an alloy had completely melted into alloy liquid, it was subject to heat preservation for 2 min, and then ejected onto the copper roller. The quenching rate of melt spinning was controlled by the linear speed of the copper wheel. A wheel speed of 20 m/s was applied in this work, to prepare melt-spun ribbons with a nanocrystalline structure.

The phase constitution of the samples in powder form was characterized by X-ray diffraction (XRD, D8 Advance, BRUKER, Karlsruhe, Germany) with Cu $K_\alpha$ radiation ($\lambda$ = 1.5418 Å, 40 kV, 40 mA). The phase analysis was performed using the Rietveld refinement with GSAS software. The magnetic properties of the ribbons were measured by a Physical Property Measurement System (PPMS, EC-II, Quantum Design, San Diego, CA, USA) equipped with a vibrating sample magnetometer (VSM) at a maximum magnetic field of 50 kOe. For the magnetic measurements, the melt-spun ribbons were cut into small pieces with a length of ∼5 mm and a width of ∼2 mm and measured in-plane to eliminate the demagnetization effect. The microstructures were characterized by transmission electron microscopy (TEM, Tecnai G2 F20 S-TWIN, Thermo Fisher Scientific Inc., Waltham, MA,

USA) equipped with an energy dispersive spectrometer (EDS), and the specimens for TEM observation were prepared by ion milling (691, Gatan, Philadelphia, PA, USA). Ion-beam thinning was carried out from the two sides of the ribbons at an inclination angle of 8° between the beam and the specimen surface. The Nd-$L_\alpha$, Pr-$L_\alpha$, Ho-$M_\alpha$ and Fe-$K_\alpha$ in the EDS spectrum were selected for mapping. Average grain size and grain-size distributions were calculated by measuring the maximum diameter with the software (Nano Measurer 1.2) for *N* grains (*N* typically in the range of 100–200).

## 3. Results and discussion

### 3.1. Phase Constitution

The XRD patterns of the melt-spun NPHFBM alloys are presented in Figure 1a. The alloys with x = 0–0.3 are only composed of hard magnetic $RE_2Fe_{14}B$ (i.e., 2:14:1) phases with the tetragonal structure (space group $P4_2/mnm$). However, an additional $REFe_2$ (i.e., 1:2) Laves phase with the Cubic structure (space group $Fd\bar{3}m$) was detected with further increasing Ho substitution (x ≥ 0.4). $HoFe_2$ phase exhibits a Curie temperature $T_c$ of 612 K, and thus it is ferromagnetic at room temperature. It can be speculated that the formation of this 1:2 phase would not affect the magnetization and remanence. However, the precipitation of the 1:2 phase would consume excess rare earth, resulting in the reduction of the non-magnetic RE-rich grain boundary phase. The RE-rich phase, such as Nd(*dhcp*), $Nd_2O_3$($P\bar{3}m$1), may exist in these alloys, but it is very difficult to distinguish by XRD, due to its low content and/or its diffraction peaks overlapping. The enlarged XRD patterns (Figure 1b) within the 2θ range of 40–44° show that the characteristic diffraction peaks of the 2:14:1 phase monotonically shift to the higher angle with increasing Ho content, indicating the lattice contraction of the 2:14:1 phase. The refined lattice parameters *a* and *c* of the 2:14:1 phase are shown in Figure 1c. The *a* and *c* of the 2:14:1 phase for the Ho-free (x = 0) alloy are 8.78(9) Å and 12.22(6) Å, respectively, which monotonically decrease to 8.73(6) Å and 12.05(2) Å, respectively, with increasing x to 0.6. Ho exhibits a smaller atomic radius of 1.77 Å than that of Nd (1.83Å) and Pr (1.82 Å). Therefore, the introduction of Ho atoms into the 2:14:1 lattice would lead to a decrease in lattice parameters, resulting in lattice contraction. In addition, the diffraction peaks of the 1:2 phase also shift to a higher angle with increasing Ho content (see Figure 1b), indicating an excess of Ho atoms have entered the 1:2 lattice, which is not beneficial for the coercivity of the alloy, and also worsens the remanence. For the x = 0.4, 0.5 and 0.6 alloys, the proportion of 1:2 phase is 2.77 wt.%, 11.09 wt.% and 22.27 wt.%, respectively.

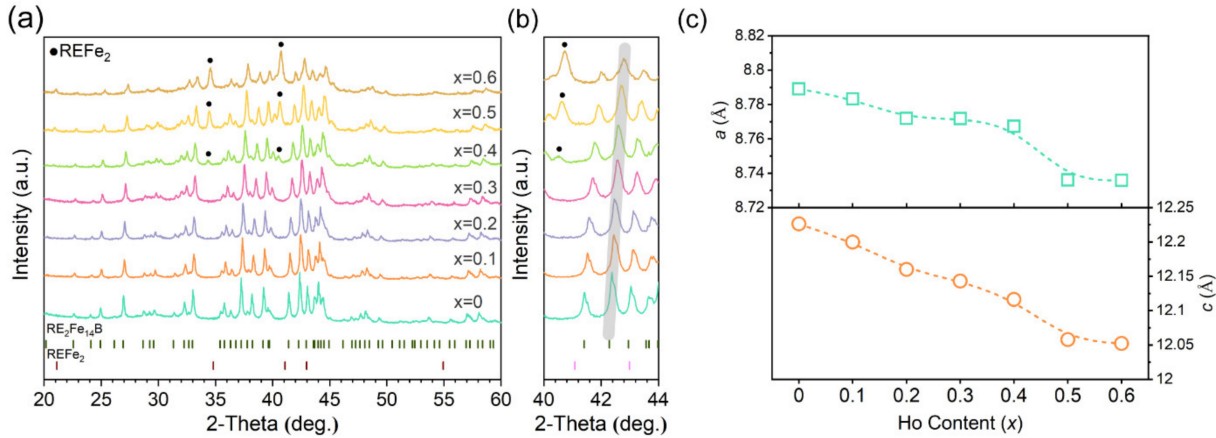

**Figure 1.** Phase structures of the NPHFBM alloys. (**a**) XRD patterns. (**b**) The enlarged XRD patterns within the 2θ range of 40–44°. (**c**) The refined lattice parameters of 2:14:1 phase.

### 3.2. Magnetic Properties

Figure 2 shows the second quadrant demagnetization curves and magnetic properties of NPHFBM alloys at 300 K. As shown in Figure 2a, the demagnetization curves of all

samples present relatively good loop squareness, indicating that uniform microstructures were obtained in all alloys. No other soft magnetic phases are presented, which is also consistent with the XRD results. The intrinsic coercivity $H_{cj}$ greatly increases from 21.1 kOe to 26.7 kOe as the doping amount (x) of Ho increases from 0 to 0.6. For the x ≥ 0.4 alloys, the precipitation of ferromagnetic 1:2 phase is generally considered not conducive to the magnetic decoupling of the 2:14:1 main phase. However, the coercivity presents a monotonically increasing trend, which should be attributed to the higher anisotropy field $H_A$ of the $Ho_2Fe_{14}B$ compound than that of $Nd_2Fe_{14}B$. The remanent magnetization $M_r$ dramatically decreases from 80.7 emu/g to 29.0 emu/g with increasing x from 0 to 0.6, which could be explained by the antiferromagnetic coupling between Ho and Fe atoms. In addition, the increasing Ho content reduces the total magnetic moment per unit volume of the 2:14:1 phase, resulting in the decreasing of both saturation magnetization Ms and Mr. Here, it should be noted that the gradual growth of the coercivity shows a slowing trend compared to the linear variation of the remanence. Combined with the XRD results, it could be concluded that it is the result of the formation of the 1:2 phase instead of the 2:14:1 phase. This means that a reasonable control of Ho doping is essential to improve the overall performance of the alloy. Remarkably, the x = 0.3 alloy shows a relatively high $H_{cj}$ of 23.9 kOe with an acceptable $M_r$ of 55.45 emu/g, which can realize the admirable maximum magnetic energy product $(BH)_{max}$ of 48 kJ/m3.

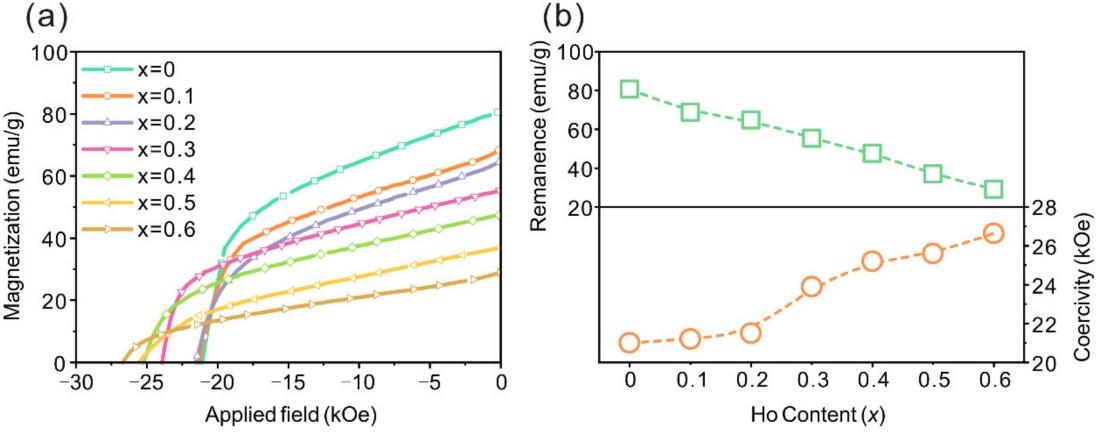

**Figure 2.** (**a**) Demagnetization curves and (**b**) the corresponding magnetic properties of the NPHFBM alloys at 300 K.

To fully understand the elevated temperature behavior of Ho-substituted NPHFBM alloys, the demagnetization curves of selected three alloys (x = 0, 0.3, 0.6) measured at different temperatures from 300 K to 400 K are shown in Figure 3a. The corresponding magnetic properties are presented in Figure 3b,c, respectively. As shown in Figure 3b, all selected samples show a monotonous decreasing trend of coercivity with increasing temperature from 300 K to 400 K. For the $RE_2Fe_{14}B$ compound, the exchange interaction of both RE-Fe and Fe-Fe becomes weaker with rising temperature, especially approaching the Curie temperature $T_c$, which also results in the decrease of $H_A$. Consequently, the $H_{cj}$ decreases with increasing temperature. In addition, the x = 0.6 alloy retains 15.5 kOe of $H_{cj}$ at 400 K, which is much higher than 11.1 kOe of the Ho-free (x = 0) alloy. It indicates that the Ho-substituted samples present higher resistance to thermal demagnetization.

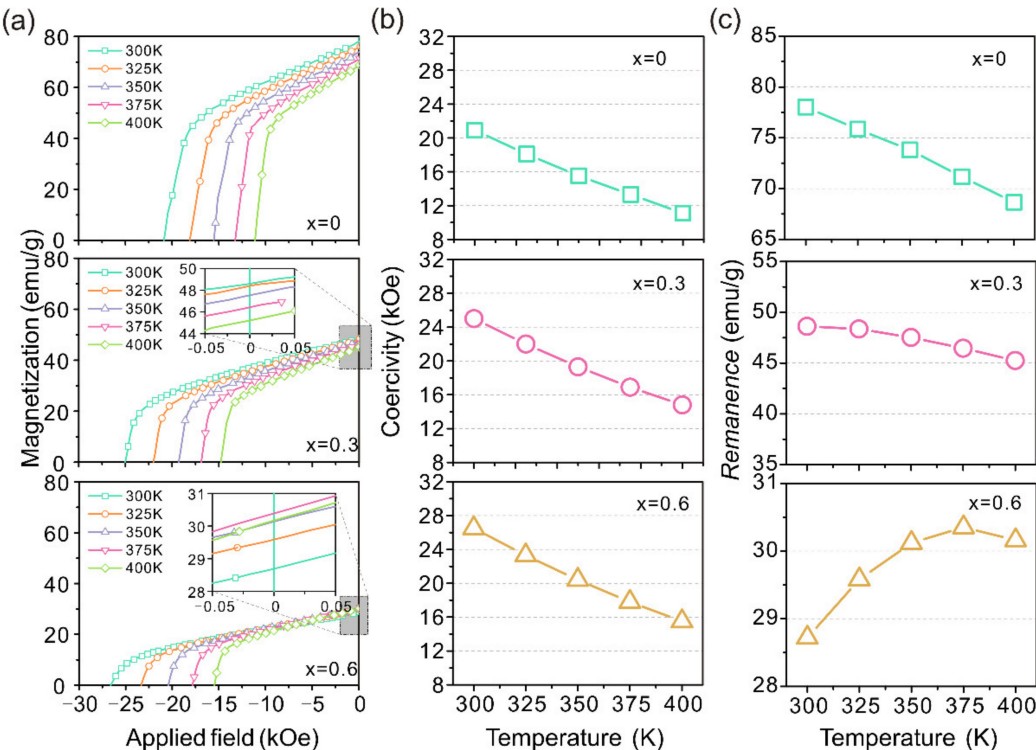

**Figure 3.** Temperature-dependent magnetic properties of NPHFBM alloys with x = 0, 0.3 and 0.6. (**a**) Demagnetization curves. (**b**) Coercivity. (**c**) Remanence.

Figure 3c shows the temperature dependences of $M_r$ for x = 0, 0.3 and 0.6 alloys, respectively. The $M_r$ for the Ho-free (x = 0) alloy decreases with increasing temperature, while the x = 0.3 alloy presents a slower downward trend. Interestingly, the $M_r$ for the x = 0.6 alloy with high Ho content slightly increases from 28.7 emu/g to 30.4 emu/g as the temperature rises from 300 K to 375 K, and then decreases to 30.2 emu/g at 400 K. This abnormal behavior is related to the temperature-dependent magnetization $M_s$ of the $Ho_2Fe_{14}B$ compound. As reported, the total magnetic moment ($M_{tot}$) of $RE_2Fe_{14}B$ is related to the coupled mode between the total RE moment ($\mu^{RE}$) and Fe moment ($\mu^{Fe}$) [18]. For $RE_2Fe_{14}B$ based on light RE elements, in which the RE moment is coupled parallel to the Fe moment, the total magnetic moment of $RE_2Fe_{14}B$ can be ascribed as $M_{tot} = 14\mu^{Fe} + 2\mu^{RE}$. However, for the heavy RE-based compound, antiparallel coupling occurs between the RE moment and Fe moment, which can be expressed as $M_{tot} = 14\mu^{Fe} - 2\mu^{RE}$. The spin order decreases with increasing temperature, resulting in the decrease of both $\mu^{Fe}$ and $\mu^{RE}$. Since the reduction of the total RE moment exceeds the Fe moment, the total magnetic moment enhancement with increasing temperature can be expected, thus resulting in a slower decrease in the remanence of the alloy, or even an unexpected increase in the remanent magnetization of the alloy in a certain temperature range. However, the remanence still appears to inevitably deteriorate as the temperature increases to 400 K. This is due to the sharp decrease of the spin order at higher temperatures, resulting in the decrease of the total magnetic moment of $RE_2Fe_{14}B$.

Generally, the temperature coefficients $\alpha$ of remanence and $\beta$ of coercivity are employed to describe the thermal stability of permanent magnetic materials, which can be defined as follows:

$$\alpha = \frac{M_r(T_2) - M_r(T_1)}{M_r(T_1) \times (T_2 - T_1)} \tag{1}$$

$$\beta = \frac{H_{cj}(T_2) - H_{cj}(T_1)}{H_{cj}(T_1) \times (T_2 - T_1)} \tag{2}$$

where $T_1$ is the initial temperature and $T_2$ is the final temperature [19]. In this work, 300 K and 400 K have been chosen as initial and final temperatures, respectively. Figure 4 presents the comparison of temperature coefficients $\alpha$ and $\beta$ for the Ho-substituted Nd-Pr-Fe-B alloys obtained in the current work with a variety of previously reported nanocrystalline melt-spun Nd-Fe-B based alloys and commercial sintered magnets at 300–400 K. As shown by the arrow, both $\alpha$ and $\beta$ values of the alloys obtained in this work are improved with increasing Ho doping, indicating the enhanced thermal stability of alloys with Ho substituting. Remarkably, a positive $\alpha$ value of 0.050 %/K is achieved in the x = 0.6 alloy. As is well known, the $\alpha$ value of $Nd_2Fe_{14}B$-type magnets is always negative, caused by the inverse relationship between $M_s$ and temperature. For commercial low-end and high-end sintered Nd-Fe-B magnets, the $\alpha$ range is from $-0.125$ %/K to $-0.75$%/K at 300–400 K, which is slightly higher the than $-0.15 \sim -0.1$ %/K of reported nanocrystalline melt-spun Nd-Fe-B ribbons [12,16–18]. In addition, the Ho-substituted Nd-Fe-B alloys obtained in this work exhibit slightly higher $\alpha$ and $\beta$ values compared with sintered Nd-Fe-B magnets, even the EH grade magnets (operating below 200 °C) with a coercivity of 30 kOe. Therefore, it suggests that the introduction of Ho into $Nd_2Fe_{14}B$-type magnets is beneficial for the magnetic properties at high temperature, especially for the $M_r$. This also provides a novel approach to improve the thermal stability of $Nd_2Fe_{14}B$-type magnets.

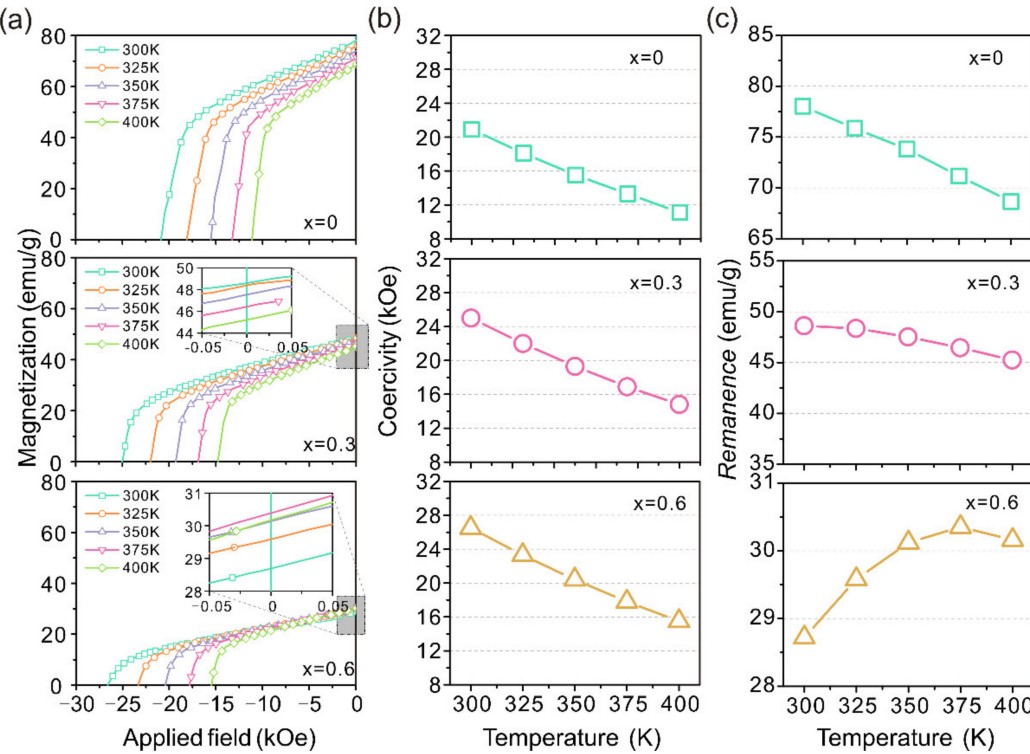

**Figure 4.** The comparison of temperature coefficients $\alpha$ and $\beta$ for the Ho-substituted Nd-Fe-B alloys obtained in the current work with a variety of previously reported nanocrystalline melt-spun (ref. from [12,20,21]) and sintered Nd-Fe-B magnets (ref. from [22]) at 300–400 K. The N-, M-, H-, SH-, UH- and EH- stand for commercial sintered magnets with different coercivity ($\geq$12 kOe, $\geq$14 kOe, $\geq$17 kOe, $\geq$20 kOe, $\geq$25 kOe and $\geq$30 kOe), which is suitable for different operating temperatures.

### 3.3. Microstructure

Figure 5a–c shows the bright-field TEM images for the selected three alloys, i.e., x = 0, 0.3 and 0.6 alloys, respectively. The corresponding grain size distributions are presented in the inset. The well-crystallized grains have been observed in all samples. It is interesting to note that the grain sizes obtained at the same speed decrease as the amount of Ho doping increases. The Ho-free sample shows a larger mean grain size of ~72.79 nm compared with x = 0.3 alloy (~46.31 nm) and x = 0.6 alloy (~33.66 nm), which implies that Ho substitution

modifies the crystallization process, which in turn affects the grain size of the alloy. As is well known, the $H_{cj}$ for Nd-Fe-B type magnets is closely related to the grain size, and higher $H_{cj}$ can be achieved with smaller grains [3]. The substitution of Ho for Nd/Pr can effectively refine the grains, which is beneficial for the $H_{cj}$. This means that the large enhancement of the coercivity is not only due to the improvement of the anisotropic field of $Ho_2Fe_{14}B$, but is also influenced by the grain size.

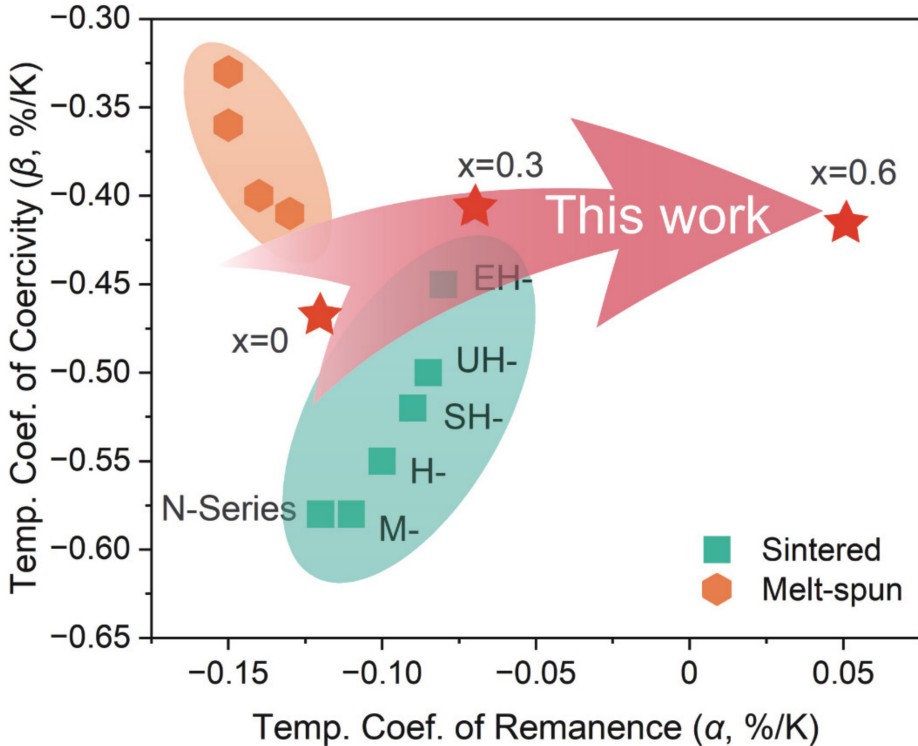

**Figure 5.** Microstructure characterization of NPHFBM alloys. Bright field TEM images and grain size distribution of the x = 0 (**a**), 0.3 (**b**) and 0.6 (**c**) alloys. The HRTEM image of the typical grain obtained from (**b**) for x = 0.3 alloy is shown in (**d**), and the corresponding FFT pattern is presented in the inset. The HAADF image obtained from (**b**) and corresponding EDS elemental mapping are shown in (**e**) and (**f**), respectively.

Figure 5d shows a high-resolution TEM (HRTEM) image of selected grains in Figure 5b for x = 0.3 alloy, and the corresponding fast Fourier transformation (FFT) pattern is shown as an inset in Figure 5d, confirming the grain is the 2:14:1 phase. A high-angle annular dark-field (HAADF) image obtained from Figure 5b is presented in Figure 5e, and the thin grain boundary layer with 1–2 nm is observed clearly. The GB layer plays an important role in $H_{cj}$ improvement, in which the GB layer can isolate the magnetic coupling between the neighboring ferromagnetic 2:14:1 grains. Figure 5f presents the EDS elemental mappings taken from Figure 5e. It shows that all the RE elements are enriched in the GB layer, and Fe is mainly distributed in the 2:14:1 grains. In addition, the distribution of Ho in the GB layer is not as obvious as that of Nd and Pr, indicating the weak segregation of Ho. This is consistent with previous results, which indicated that the light RE prefers to enter into the GB compared with the heavy RE [23,24].

## 4. Conclusions

In this work, we systematically studied the effects of Ho addition on the phase constitution, magnetic properties and microstructures of nanocrystalline melt-spun $[(Nd_{0.8}Pr_{0.2})_{1-x}Ho_x]_{14.3}Fe_{76.9}B_{5.9}M_{2.9}$ (M=Co, Cu, Al and Ga) (at. %; x = 0–0.6) alloys. The REFe$_2$ phase is formed at high amounts of Ho substitution (x ≥ 0.4). At room temperature, the coercivity greatly improved from 21 kOe to 27 kOe with increasing Ho substitution from 0 to 60%, while the

remanent magnetization deteriorated from 80.7 emu/g to 29.0 emu/g accordingly. For the elevated temperature behavior, the $M_r$ of the x = 0.6 alloy increases with rising temperature from 300 K to 375 K, which is attributed to the antiparallel coupling between Ho and Fe moments. Both the temperature coefficient $\alpha$ of $M_r$ and $\beta$ of $H_{cj}$ are improved by Ho substitution, indicating the enhanced thermal stability by Ho addition. Remarkably, the positive value of $\alpha$ = 0.050%/K is achieved in x = 0.6 alloy (300–400 K), which is superior to that of Nd-Fe-B magnets. The TEM results have revealed that Ho substitution can refine the size of the 2:14:1 grain. The EDS elemental mapping shows the RE segregation, in which Nd and Pr are more likely to segregate in the grain boundary phase than Ho. The present work provides a practical road map for enhancing the coercivity and thermal stability of Nd-Fe-B permanent magnetic materials simultaneously.

**Author Contributions:** Conceptualization, C.X. and X.L.; methodology, C.X.; software, W.Z.; validation, C.X., W.Z. and Y.T.; formal analysis, C.L.; investigation, C.X.; writing—original draft preparation, C.X.; writing—review and editing, X.L. and Q.Z..; visualization, C.L. and Y.T.; supervision, Z.Z.; funding acquisition, R.T. and Q.Z. All authors have read and agreed to the published version of the manuscript.

**Funding:** This work was supported by the Guangdong Basic and Applied Basic Research Foundation, China (No. 2022A1515011453, No. 2021A1515010800), and the GDAS Project of Science and Technology Development (No. 2019GDASYL-0103067, No.2022GDASZH-2022010104, No. 2022GDASZH-2022030604-04).

**Conflicts of Interest:** The authors declare no conflict of interest.

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
