# Peer review of "Effect of Ho Substitution on Magnetic Properties and Microstructure of Nanocrystalline Nd-Pr-Fe-B Alloys"

_metals, doi:10.3390/met12111922_

Round 1
Reviewer 1 Report
This paper focuses on improving the magnetic properties of RE-Fe-B alloys by substituting the Nd and Pr elements with Ho. The best results are obtained for a substitution rate of 33%. The work seems well done but calls for several remarks which must be taken into consideration with the acceptance of the article. 1- The role of 3d alloying elements (Co, Cu, Al, Ga) must be specified 2- Specify the temperature and the time of elaboration of the alloys 3- Indicate the obtained proportion of the secondary phase HoFe2 4- HoFe2 is ferromagnetic at room temperature. It must therefore have an influence on the coercive field of the alloy. This point is not discussed in the submitted version. 5- Reference 12 is not complete. Journal name is missing
Author Response
Responses to the reviewer 1’ comments:
This paper focuses on improving the magnetic properties of RE-Fe-B alloys by substituting the Nd and Pr elements with Ho. The best results are obtained for a substitution rate of 33%. The work seems well done but calls for several remarks which must be taken into consideration with the acceptance of the article.
1-The role of 3d alloying elements (Co, Cu, Al, Ga) must be specified
Response: Thanks for review comments. We have added one paragraph at the end of Introduction section to clarify the role of 3d alloying elements (Co, Cu, Al, Ga) in Nd-Fe-B magnets, as follows.
“In addition, the introduction of trace elements has a positive effect on the magnetic properties of Nd-Fe-B magnets. Substituting Co element for Fe can significantly increase the Curie temperature (Tc) of Nd2Fe14B phase, thus Co is usually employed to improve the thermal stability of Nd-Fe-B magnets [14]. Although Al is not favorable to Tc, it is found to be beneficial for coercivity by partially substituting Fe for a decrease in domain wall thickness [15]. Adding Ga and Cu can regulate the grain boundary and results in coercivity enhancement [16, 17].”
2- Specify the temperature and the time of elaboration of the alloys
Response: We have added a paragraph to make it clear in the Experimental to make it clear as follows: “The nanocrystalline melt-spun ribbons were prepared by melt spinning. The alloys with a total mass of 100 g was melted by induction melting, the general steps of alloy melting were as follows: 1 kW heat preservation for 1 min for preheating, then in-crease the power to 7 kW for melting, after it is completely melted into alloy liquid, heat preservation for 2 min, and then ejected onto the copper roller.” The processing temperature depends on the melting point of the alloys.
3- Indicate the obtained proportion of the secondary phase HoFe2
Response: We have added a paragraph to make it clear in the Results and Discussion to make it clear as follow: “For the x=0.4, 0.5 and 0.6 alloys, the proportion of 1:2 phase is 2.77 wt.%, 11.09 wt.% and 22.27 wt.%, respectively.”
4- HoFe2 is ferromagnetic at room temperature. It must therefore have an influence on the coercive field of the alloy. This point is not discussed in the submitted version.
Response: We have added a paragraph to make it clear in the manuscript to make it clear as follow: “The intrinsic coercivity Hcj greatly increases from 21.1 kOe to 26.7 kOe as the doping amount (x) of Ho increases from 0 to 0.6. For the x≥0.4 alloys, the ferromagnetic 1:2 phase precipitate, which is generally considered not conducive to magnetically decoupling of 2:14:1 main phase. However, the coercivity presents a monotonically increasing trend, which should be attributed to the higher anisotropy field HA of Ho2Fe14B compound than that of Nd2Fe14B.”
5- Reference 12 is not complete. Journal name is missing
Response: Sorry for the mistake, we had completed the reference 12 as follow: “D.N. Brown, D. Lau, Z. Chen, Substitution of Nd with other rare earth elements in melt spun Nd2Fe14B magnets, AIP Advances, 6 (2016) 056019.”
Reviewer 2 Report
The presented article is interesting, thematically suitable for journal Metals, but it has a lot of confusing and poorly explained information for the reader, so it is necessary to correct some statements in the text and pictures, too.
There are different types of different fonts throughout the text, it is necessary to unify it, e.g. lines 123, 124 or 139, 140, but also more.
There are missing spaces before parentheses throughout the text. For example spaces are missing everywhere before the parenthesis that introduces the citation.
See next:
· Line 48: What does it mean "can improve coercivity"? It is necessary to explain the coercivity relation for hard magnetic materials. Respectively, the difference in values of coercivity for soft and hard magnetic materials. And it is also necessary to provide relevant citations.
· Lines 16, 47, 49, 58: Coercivity of feromagnetic material is the intensity of the applied magnetic field required to demagnetize the material. There are different definitions of coercivity depending on what is considered to be "demagnetized". What kind of coercivity is referred to in the preseneted paper (probably the intrinsic coercivity). This needs to be adjusted, because it is confusing for reader and it is definitelly necessary to quote A/m (common units for coercivity) and T units when and how they should be given correctly. And it is also necessary to provide relevant citations.
· Figures 2b, 3b: Units of coercivity are not correct
· Line 182: Formulas are in not correct format and without number in text. And not all quantities are mentioned and marked in the text
· Figure 4: What does it mean this objects inside figure? It is necessary to explain it or delete it.
· Figure 5: The quality of photos a, b, c is no tok, it is necessary to chcage it or to more explain in text.
· Conclusion: The conclusion is poorly elaborated, some facts are mentioned for the first time, e.g. phase in line 242, it needs to be reworked.

Author Response
The presented article is interesting, thematically suitable for journal Metals, but it has a lot of confusing and poorly explained information for the reader, so it is necessary to correct some statements in the text and pictures, too.
There are different types of different fonts throughout the text, it is necessary to unify it, e.g. lines 123, 124 or 139, 140, but also more.
Response: We had unified the font and checked the format problems.
There are missing spaces before parentheses throughout the text. For example spaces are missing everywhere before the parenthesis that introduces the citation.
Response: We had added spaces before parentheses, and fully checked all text.
See next:
- Line 48: What does it mean "can improve coercivity"? It is necessary to explain the coercivity relation for hard magnetic materials. Respectively, the difference in values of coercivity for soft and hard magnetic materials. And it is also necessary to provide relevant citations.
Response: The coercivity is depends on the anisotropy field HA of the ferromagnetic main phase, although Ho2Fe14B only shows a slightly higher HA than that of Nd2Fe14B, the introduction of Ho into Nd-Fe-B magnets can dramatically improve the coercivity. So we modified the paragraph to make it clear as follows: “Although Ho2Fe14B only presents a slightly higher HA (75 kOe) than 73 kOe of Nd2Fe14B, several researchers indicated that introducing Ho into Nd-Fe-B magnets can dramatically improve the coercivity.”
Hard magnetic materials and soft magnetic materials are mainly different in the value of coercivity. Soft magnetic materials typically exhibit coercivities values of approximately 400 A m-1 (5 Oe) to as low as 0.16 A m-1 (0.002 Oe). Soft magnetic behavior is important in any application involving a change in magnetic induction. Hard magnetic materials retain a large amount of residual magnetism after exposure to a magnetic field. These materials typically have coercivities, Hc, of 10 kA/m (125 Oe) to 1 MA/m (12 kOe). The materials at the high coercivity end of this range are known as permanent magnets. These materials are used principally to supply a magnetic field [Ref.: D.C. Jiles, Recent advances and future directions in magnetic materials, Acta Materialia, 51 (2003) 5907–5939].
Lines 16, 47, 49, 58: Coercivity of feromagnetic material is the intensity of the applied magnetic field required to demagnetize the material. There are different definitions of coercivity depending on what is considered to be "demagnetized". What kind of coercivity is referred to in the preseneted paper (probably the intrinsic coercivity). This needs to be adjusted, because it is confusing for reader and it is definitelly necessary to quote A/m (common units for coercivity) and T units when and how they should be given correctly. And it is also necessary to provide relevant citations.
Response: When the reverse magnetic field H increases to a certain value Hcj, the vector sum of the micro-magnetic dipole moment inside the magnet is 0, and the reverse magnetic field H value is called the intrinsic coercive force Hcj of the material. Therefore, the coercivity in this paper is intrinsic coercivity, so its unit is changed to the Gauss unit system kOe of intrinsic coercivity [Ref.: G P Zhao, L Zhao, L C Shen, Coercivity mechanisms in nanostructured permanent magnets, Chinese Physics B, 28 (2019) 077505].
Figures 2b, 3b: Units of coercivity are not correct
Response: We had unified the units to Gauss unit system. The unit of coercivity have been changed from “T” as “kOe”.
Line 182: Formulas are in not correct format and without number in text. And not all quantities are mentioned and marked in the text
Response: We had added the formulas number, and explained all the quantities
Figure 4: What does it mean this objects inside figure? It is necessary to explain it or delete it.
Response:To make it clear, we had moved the text of “this work” into the arrow inside. We added a sentence to make it clear, as follows: “As shown by the arrow, both α and β values of the alloys obtained in this work are improved with increasing Ho doping, indicating the enhanced thermal stability of alloys with Ho substituting.” It means the samples were obtained in this work and with the increase of Ho content, its thermal stability has been improved.
Figure 5: The quality of photos a, b, c is not ok, it is necessary to chcage it or to more explain in text.
Response: We had adjusted the resolution and brightness of Figs. 5(a-c) and enlarged the inset figure.
Conclusion: The conclusion is poorly elaborated, some facts are mentioned for the first time, e.g. phase in line 242, it needs to be reworked.
Response: We changed the “REFe2 phase” to “1:2 phase”. And we had explained this phase structure in the “3.1. Phase constitution” part, as follow: “However, an additional REFe2 (i.e. 1:2) Laves phase with the Cubic structure (space group Fd m) was detected with further increasing Ho substitution (x≥0.4).”
Reviewer 3 Report
As a practical road map for enhancing the coercivity and thermal stability of Nd-Fe-B permanent magnetic materials, it is a good work, but I have doubts whether the 5T maximum magnetic field is sufficient for all premagnetization processes to take place for the presented coercivity values. With regard to the trapping of domain walls (the so-called pinning effect), especially on such an inhomogeneous microstructure as presented in the article, this can significantly affect the resulting coercivity of the measured sample. It would be appropriate to supplement the presented data with a presentation of the full hysteresis loop of these materials. It is understandable that the authors are limited by the experimental possibilities of the device, but this fact (effect of maximum magnetic field) should be commented in the text.
Technical comment:
Row 99: it is Laves not Lave phase!
In image three, it would be necessary to unify the font type and size for the axis descriptions
Round 2
Reviewer 1 Report
The authors have taken into account the comments and suggestionsmade in the revised version.
The article is now acceptable for publication in Metals
Reviewer 2 Report
I agree with the modification after review.